# Isolation, Characterization and Bioactive Properties of Alkali-Extracted Polysaccharides from *Enteromorpha prolifera*

**DOI:** 10.3390/md18110552

**Published:** 2020-11-06

**Authors:** Shifeng Zhao, Yuan He, Chungu Wang, Israa Assani, Peilei Hou, Yan Feng, Juanjuan Yang, Yehua Wang, Zhixin Liao, Songdong Shen

**Affiliations:** 1Department of Pharmaceutical Engineering, School of Chemistry and Chemical Engineering, Southeast University, Nanjing 211189, China; shifengzhao@seu.edu.cn (S.Z.); chunguwang404@seu.edu.cn (C.W.); israaassani@seu.edu.cn (I.A.); peileihou@seu.edu.cn (P.H.); yanfeng@seu.edu.cn (Y.F.); 2Jiangsu Province Hi-Tech Key Laboratory for Bio-Medical Research, Southeast University, Nanjing 211189, China; 3Department of Cell Biology, School of Biology and Basic Medical, Soochow University, No. 199 Renai Road, Suzhou 215123, China; heyuan@suda.edu.cn (Y.H.); jjyang2018@stu.suda.edu.cn (J.Y.); yhwang@stu.suda.edu.cn (Y.W.)

**Keywords:** *Enteromorpha prolifera*, characterization, antioxidant activity, antitumor activity

## Abstract

Four new purified polysaccharides (PAP) were isolated and purified from the *Enteromorpha prolifera* by alkali extraction, and further characterization was investigated. Their average molecular weights of PAP-1, PAP-2, PAP-3, and PAP-4 were estimated as 3.44 × 10^4^, 6.42 × 10^4^, 1.20 × 10^5^, and 4.82 × 10^4^ Da, respectively. The results from monosaccharide analysis indicated that PAP-1, PAP-2, PAP-3 were acidic polysaccharides and PAP-4 was a neutral polysaccharide. PAP-1 and PAP-2 mainly consist of galacturonic acid, while PAP-3 and PAP-4 mainly contained rhamnose. Congo red test showed that no triple helical structure was detected in the four polysaccharides. The antioxidant activities were investigated using 1,1-diphenyl-2-picrylhydrazyl (DPPH), Superoxide, and 2, 2′-azino-bis(3-ethylbenzothiazoline-6-sulfonic acid) (ABTS) radical assay. In vitro antitumor activities were evaluated by 3-(4,5-Dimethylthiazol-2-yl)-2,5-diphenyltetrazolium bromide (MTT) assay. PAP-1 exhibited relatively stronger antioxidant activities among the four polysaccharides in a dose-dependent manner. At a concentration of 1.00 mg/mL, the antioxidant activities of PAP-1 on the DPPH radical scavenging rate, superoxide anion radical scavenging rate, and ABTS radical rate at 1.00 mg/mL were 56.40%, 54.27%, and 42.07%, respectively. They also showed no significant inhibitory activity against MGC-803, HepG2, T24, and Bel-7402 cells. These investigations of polysaccharides provide a scientific basis for the use of *E. prolifera* as an ingredient in functional foods and medicines.

## 1. Introduction

*Enteromorpha prolifera* is an annual green alga and belongs to the genus Enteromorpha which is widely distributed in the inter-tidal zone of the ocean, particularly in the eastern regions of China. This algae is rich in polysaccharides, proteins, minerals, and vitamins [1]. In ancient Chinese culture, it has been used as a food source and traditional medicine to treat many diseases.

*Enteromorpha* exhibits a strong propagation capability. In recent decades, large-scale green tides in the Yellow Sea, China, have negative impacts on the coastal areas [2]. In 2008, the largest green algae bloom was reported in the coastal region of Qingdao and covered an area of approximately 13,000–30,000 km^2^, with *Enteromorpha* as the main species [3]. The decomposition of *E. prolifera* often causes unsightly appearance, foul odors, and adverse public reactions.

*Enteromorpha* species contain large amounts of polysaccharides [4]. Researchers have found that polysaccharides from *Enteromorpha* possess a wide range of pharmacological perspectives, such as antitumor, antioxidant, and hypolipidemic properties [5,6,7]. Extraction is the first and essential step for the characterization and utilization of bioactive polysaccharides from plants [8]. Processing conditions can change the structure and composition of polysaccharides, providing both desirable and undesirable effects on their functional properties [9,10]. Meanwhile, researchers have reported that alkaline extraction is an efficient method to obtain extracts rich in bioactive polysaccharides [11,12,13]. In recent years, most relevant studies have evaluated the chemical composition and biological activities of polysaccharides isolated from *E. prolifera* using hot water extraction. However, there were no reports published about the characterization, antioxidant activities of polysaccharides obtained by alkaline extraction from *E. prolifera* until now.

Hence, in the present work four new polysaccharides were isolated from *E. prolifera* using alkaline extraction. Then, their structures were analyzed by high-performance gel permeation chromatography (HPGPC), high-performance liquid chromatography (HPLC), infrared spectroscopy (FT-IR), Congo red test, and scanning electron microscopy (SEM) analysis. Moreover, the antioxidant and antitumor activities were also evaluated in vitro. These results could be used for future investigation of the structure-activity relationship and development of application of *E. prolifera* polysaccharides.

## 2. Results and Discussion

### 2.1. Isolation and Purifiction of Polysaccharides

Using DEAE-52 cellulose ion-exchange chromatography with a stepwise concentration gradient of NaCl solution (0, 0.3, 0.5, 0.7 M), crude polysaccharide (AP) was separated into four fractions (Figure 1), named as AP-1, AP-2, AP-3, and AP-4, respectively. Each of the four fractions was further purified with Sephadex G-100 column chromatography using distilled water as an eluent. The purified polysaccharides PAP (PAP-1, PAP-2, PAP-3, and PAP-4) were obtained, respectively. The gel filtration chromatograms of the four fractions are reported in Figure 2.

### 2.2. Determination of Molecular Weight

The HPGPC elution profiles showed of each polysaccharide has a single and symmetrically sharp peaks, which indicated the polysaccharides were homogeneous. According to the retention times, the average molecular weights of PAP-1, PAP-2, PAP-3, and PAP-4 were 3.44 × 10^4^, 6.42 × 10^4^, 1.20 × 10^5^, and 4.82 × 10^4^ Da, respectively (Figure 3).

### 2.3. Monosaccharides Composition

The monosaccharide composition of purified polysaccharides was analyzed by HPLC. The results are shown in Table 1 and Figure 4. PAP-1 was composed of eight monosaccharides, and PAP-2 was composed of five monosaccharides, both were mainly composed of galacturonic acid. The main component of PAP-3 and PAP-4 was rhamnose. As can be seen, galacturonic acid was presented in all purified polysaccharides except for PAP-4, indicating that PAP-1, PAP-2, and PAP-3 were acidic polysaccharides, while PAP-4 was a neutral polysaccharide.

### 2.4. UV and Infrared Spectrum Analysis

No absorption was detected by the UV spectrum (Figure 5a) at 260 and 280 nm, indicating there is no free proteins in the samples. The FT-IR spectrum of purified polysaccharides (PAP-1, PAP-2, PAP-3, and PAP-4) are shown in Figure 5b. The strong and broad absorption peak at 3430 cm^−1^ was attributed to the stretching vibrations of O-H groups, and a weak absorption at 2970 cm^−1^ was corresponded to C-H stretching vibrations. The absorption peaks which observed at approximately 1630 and 1400 cm^−1^ were assigned to the asymmetric and symmetric stretching of the carboxylate anions groups [14]. The weak absorption peak at 1250 cm^−1^ was assigned to the stretching vibrations of S-O’s stretching vibration, indicating that sulfation groups (SO_3_)^−^ might be included [15]. The absorption peaks at 1050 and 1080 cm^−1^ were equivalent to the corresponding to C–O–C glycosidic band and expansion vibration of an asymmetric ring of pyranose ring. The result suggested the presence of a pyranose ring [14]. The presence of two characteristic peaks at about 850 and 899 cm^−1^ indicated that PAP-3 had both α-type and β-type glycosidic linkages [16].

### 2.5. Congo Red Assay

The triple-helix conformation of polysaccharides plays an important role in their biological activities, especially the antitumor activities, which can be identified using the Congo red test [17]. The polysaccharides with a triple-helix structure, can form a complex with Congo red under weak alkaline conditions, which will display a λ_max_ redshifted compared to the Congo red solution [18]. With the NaOH concentration gradient increased, the triple helix structure in the polysaccharide decomposed into a single strand, and the red-shift effect weakened. As shown in Figure 6, the expected trend for triple-helix polysaccharides was not observed, indicating that purified polysaccharides do not have a triple-helix structure. It can be observed that the λ_max_ of PAP-1 increased firstly and then kept constant with the increase of NaOH concentration, which indicated that PAP-1 might have a different structure.

### 2.6. SEM Analysis

SEM could be set as a characteristic to qualitatively identify the spatial structure of alkali-extracted polysaccharides. Therefore, SEM images of purified polysaccharides at a magnification of 6000 were obtained (Figure 7). The microstructure of PAP-1, PAP-2, and PAP-4 appeared as large amounts of rough-surfaced lamellar structures. They intertwined with each other irregularly to form a reticular layer, which was similar to the alkali-extracted polysaccharide from *Hericium erinaceus* [19]. The texture of PAP-3 had become rougher-surfaced with strips and granules. Notably, the surface of PAP-3 was mixed with filamentous granules. The four purified polysaccharides showed different morphologies under SEM related to their molecular weight, monosaccharide composition, and linkages. In addition, different methods of purification and elution used also had affected the surface morphology of polysaccharides.

### 2.7. Antioxidant Activities In Vitro

#### 2.7.1. DPPH Free Radical Scavenging Assay

DPPH is a stable free radical which is widely used to determine the ability of compounds to act as free radical scavengers or radical hydrogen donors and to measure the antioxidant activities, which loses its absorption spectrum band at 515–528 nm when it accepts an electron or a free radical species. Antioxidants transfer electrons or hydrogen atoms to reduce DPPH radicals’ number equal to the number of available hydroxyl groups [20]. The DPPH-Extract solution was incubated for 15 and 30 min, respectively. As shown in Figure 8a,b, under the two incubations times, the scavenging ability of various polysaccharide fractions increased with the increasing concentration of the dosage range of 0–1.00 mg/mL. The DPPH radical scavenging ability decreased in the order of ascorbic acid (V_c_) > PAP-1 > PAP-3 > PAP-4 > PAP-2. After incubating for 15 min, the DPPH free radical scavenging rate of V_c_, PAP-1, PAP-2, PAP-3, and PAP-4 were 91.30% ± 0.06, 51.60% ± 0.02, 30.67% ± 0.08, 40.33% ± 0.07, and 33.33% ± 0.01 at a concentration of 1.00 mg/mL, respectively. The scavenging rates were lower than the former rates. The former scavenging rates of V_c_, PAP-1, PAP-2, PAP-3, and PAP-4 after incubation for 30 min were 97.00% ± 0.07, 56.40% ± 0.05, 32.67% ± 0.03, 44.00% ± 0.04, 37.33% ± 0.03 at a concentration of 1.00 mg/mL, respectively. The incubation time had an effect on scavenging ability, that means the longer incubation time leads to higher scavenging rates. Besides, samples’ scavenging activities increased significantly with the increasing of concentration ranging from 0.05 to 0.20 mg/mL. The results indicated that all the samples possessed scavenging activities on DPPH, and mainly PAP-1 had more potent antioxidant activity.

#### 2.7.2. Superoxide Anion Radical Scavenging Assay

Superoxide anion plays an important role in plant tissues and is involved in the formation of other cell-damaging free radicals [21]. Superoxide anion radical decomposes to form stronger reactive oxidative species (ROS) such as hydrogen peroxide, hydroxyl radical, and singlet oxygen, which induce oxidative damage in lipids, proteins, and DNA [22]. As shown in Figure 8b, the superoxide scavenging activities of four polysaccharides were increased very significantly with concentrations in the range of 0.05–0.40 mg/mL. The superoxide radical scavenging rate of V_c_, PAP-1, PAP-2, PAP-3, and PAP-4 at the concentration of 1.00 mg/mL was 96.67% ± 0.08, 54.27% ± 0.04, 37.83% ± 0.03, 42.50% ± 0.03, and 36.92% ± 0.02, respectively. In the system of pyrogallol’s autoxidation, all samples could inhibit the autoxidation of pyrogallol. PAP-1 showed a stronger ability than others. The possible mechanism of PAP-1 scavenging superoxide radical is due to lower molecular weight and spatial structure, that they can combine with superoxide anion radical and form a stable radical to terminate the radical chain reaction.

#### 2.7.3. ABTS Scavenging Assay

The means of scavenging a protonated radical ABTS was extensively applied to evaluate the total antioxidant ability potential of natural products [23]. Concentration-dependent radical scavenging activity was observed in Figure 8c. The ABTS scavenging rate of V_c_, PAP-1, PAP-2, PAP-3 and PAP-4 at 1.00 mg/mL was 98.33% ± 0.08, 42.07% ± 0.29, 35.93% ± 0.23, 33.47% ± 0.23, and 22.07% ± 0.15, respectively. The scavenging ability of PAP-1 had the highest rate of 42.07% ± 0.29 at the concentration of 1.00 mg/mL, while the scavenging ability of PAP-4 was reduced by almost a half to PAP-1, the highest scavenging ratio was lower than 50%, and the PAP-2 also exhibited strong ability on scavenging the ABTS radical, while the PAP-4 exhibited the lowest ABTS radical scavenging effects.

#### 2.7.4. Hydrogen Peroxide Hemolysis Assay

In this study, the H_2_O_2_ induced hemolysis inhibitory activity assay was used to further evaluate antioxidant activities of the purified polysaccharides. As shown in Figure 8, the *E. prolifera* polysaccharide showed moderate inhibitory rates. As can be seen in Figure 8d, at the concentration of 1.00 mg/mL, the inhibitory rate of V_c_, PAP-1, PAP-2, PAP-3, and PAP-4 was 85.33% ± 0.05, 31.86% ± 0.22, 25.34% ± 0.18, 27.12% ± 0.19, and 19.86% ± 0.14, respectively. PAP-1 showed the highest inhibitory rate among four purified polysaccharides, but it was much less effective than V_c_.

In the report of Li [24], WE-32 exhibited stronger antioxidant activities among six polysaccharides, which were extracted from the same Ulva family of Enteromorpha intestinalis by hot water. The scavenging rate of WE-32 on DPPH and superoxide radicals was 77.13% ± 0.68 and 70.40% ± 0.48 at a concentration of 10 mg/mL. Compared with their result, the alkali-extracted polysaccharides showed higher antioxidant activity than water-extracted polysaccharides. The antioxidant activities of polysaccharides were closely related to their molecular weight, monosaccharide composition, spatial structure, polysaccharide chain branching, and even correlated to the selected antioxidant evaluation system [25]. In general, all tested samples possessed antioxidant activities in vitro assay. This paper comprehensively uses four methods from different perspectives to evaluate the antioxidant activity of *E. prolifera* polysaccharides which avoid the one-sided and insufficiency used by a single antioxidant method. The higher content of galacturonic acid in PAP-1 generally exhibited better antioxidative capacity than other samples for the in vitro assays [26,27]. Congo red assay indicated that PAP-1 might have a different structure, which might also lead to the highest scavenging rate. PAP-2 contained the highest galacturonic acid content but did not show better scavenging rates, and PAP-4 showed the minimum scavenging rate. The reason might be the tangled strip spatial structure leading to the lower solubility (data not shown) so that radicals cannot be scavenged effectively. PAP-3 also showed a better antioxidative capacity despite the highest molecular weight. The possible hypothesis for this result might be the maximum molecular weight and the relatively loose spatial structure that could effectively be combined with more radials and resulted in the second scavenging rate. Since PAP-1 could be served as a promising natural antioxidant and to be considered as a potential substitute as well. However, further studies are required to elucidate the biological mechanisms.

### 2.8. Antitumor Analysis In Vitro

In this study, the inhibitory effects of PAP-1, PAP-2, PAP-3, and PAP-4 on MGC-803 (Human gastric cancer cells), HepG2 (Human liver cancer cells), T24 (Human bladder cancer cells), Bel-7402 (Human liver cancer cells), and AG1522 (normal human foreskin fibroblast cells) cells in vitro were investigated. Doxorubicin (DOX) was used as a positive control. The result was shown in Table 2. In the preliminary screening test, the inhibition ratio of MGC-803, HepG2, T24, Bel-7402 cells induced by the four purified polysaccharides was not more than 30% at the concentration of 0.04 mg/mL, suggesting that the purified polysaccharides had no significant effects on MGC-803, HepG2, T24 and Bel-7402 cells in vitro. All polysaccharide samples have shown no cytotoxic activity on AG1522 cells. The scavenging rates of crude polysaccharide (AP) and preliminary purified polysaccharides APs are lower than those of purified polysaccharide. The reason might be that the purified polysaccharides did not contain impurities such as pigments. The higher degree of purification gives the better scavenging efficiency. It has also been reported that the antitumor activity of polysaccharides depends on their molecular weight, monosaccharide composition, and spatial structure, such as triple-helix conformation. The triple-helix conformation is an essential structural requirement for the antitumor effects of polysaccharides [28]. In this study, the Congo red assay result indicated that the purified polysaccharides do not have a triple-helix structure. This might lead to the poor antitumor inhibitory effects.

## 3. Materials and Methods

### 3.1. Materials

The *E. prolifera* were collected from the culture nets in coastal regions in Qingdao City, China. The plant was identified by Prof. Songdong Shen, School of Biology and Basic Medical, Soochow University. The DEAE-cellulose 52, Sephadex G-100, and monosaccharide standards of fucose (Fuc), rhamnose (Rha), arabinose (Ara), galactose (Gal), glucose (Glc), xylose (Xyl), mannose (Man), galacturonic acid (GalA), and glucuronic acid (GlcA) were obtained from YuanYe Bio-Technology Co. (Shanghai, China). All cancer cell lines were purchased from China Life Science College (Shanghai, China). All other chemicals used were of analytical grade and were used as received without any further purification.

### 3.2. Preparation of the Crude Polysaccharide

The *E. prolifera* powder (500 g) was defatted with 95% ethanol at 100 °C for 2 h. The resulting mixture was cooled and filtered, then extracted three times with deionized water 1:10 (*w/v*) at 80 °C for 2 h. The sediments were collected and extracted with NaOH solution (0.30 mol/L) 1:10 (*w/v*) at room temperature for 3 h and pooled (twice). Then the extracts were filtered, neutralized with acetic acid (0.30 mol/L), and concentrated. Next, 95% ethanol (*v/v* = 1:4) was added to the concentrated solution and reacted at 4 °C for 24 h to obtain crude polysaccharide, which was deproteinized by the Sevage method [29]. Finally, the resulting solution was concentrated under reduced pressure and freeze-dried to provide crude polysaccharide (AP) as a pale yellow powder.

### 3.3. Separation and Purification of AP

AP (2 g) was purified sequentially by DEAE-52 cellulose and Sephadex G-100 filtration chromatography according to the reported methods with little modification [30]. The AP was applied tardily to a column (3 × 60 cm) of DEAE-52 cellulose. The column was eluted with distilled water (500 mL), which was followed by 0.3, 0.5, 0.7 M NaCl (500 mL) at a flow rate of 1.0 mL/min. Following this, the obtained elutes (10 mL per tube) were collected by the automatic collector and monitored by phenol-sulfuric acid method at 490 nm. Finally, four fractions named AP-1, AP-2, AP-3, and AP-4 were obtained. The resulting fractions were mixed, concentrated, dialyzed against distilled water, and lyophilized. The solution of each fraction was further purified through the Sephadex G-100 column (2.5 × 60 cm). The elutes were collected (4 mL per tube) automatically. The appropriate fractions were mixed, concentrated, dialyzed, and lyophilized, resulting, with four purified polysaccharides named PAP-1 (60 mg), PAP-2 (85 mg), PAP-3 (57 mg), and PAP-4 (102 mg) for further investigation.

### 3.4. Determination of Molecular Weight

The polysaccharides were analyzed using a UV spectrophotometer (UV-2600, Shimadzu, Kyoto, Japan) to detect the presence of nucleic acids and proteins. High-performance gel permeation chromatography (HPGPC, Agilent Technologies Inc., Palo Alto, CA, USA) was used to determine the average molecular weight. The columns (PL aquagel-OH 8 µM, 7.5 × 300 mm), eluted with 0.1 M sodium nitrate solution and 500 ppm sodium azide at a flow rate of 1.0 mL/min at 35 °C [31]. Dextran standards with different molecular weights (T1, T5, T10, T70, T500, and T2000) were used to conduct the standard calibration curve.

### 3.5. FT-IR Spectroscopy Analysis

The FT-IR spectra of purified polysaccharides were measured by a spectrometer instrument (Nicolet 5700, Thermo Fisher Scientific, Waltham, MA, USA). The polysaccharides were mixed with dried potassium bromide powder and pressed to make a transparent film. The film was put into the FTIR instrument and scanned from 400 to 4000 cm^−1^.

### 3.6. Determination of Monosaccharide Composition

The pre-column 1-phenyl-3-methyl-5-pyrazolone (PMP) derivatization HPLC method was used to characterize and quantify the monosaccharide composition of polysaccharides [32]. Briefly, 5 mg of crude polysaccharides was dissolved with 2 mL of 3 M trifluoroacetic acid (TFA), the mixture was hydrolyzed in a sealed tube at 120 °C for 6 h. Methanol was added to the mixture to remove the excess of TFA, and this process was repeated 3 times. Then, 100 µL of the sample solution was dissolved in 100 µL of 0.5 M methanolic solution of PMP. After adding 100 µL of 0.3 M NaOH, the mixture was incubated for 30 min at 70 °C. After cooling down to room temperature, the mixture was acidified for neutralization by adding 105 µL of an HCl solution (0.3 M). The resulting solution was extracted three times by adding chloroform until the organic layer was carefully removed. The sample was passed through a column (Phenomenex Gemini C18, 5 μM, Ø 4.6 × 250 mm) at 25 °C and eluted with a mixture of 0.1 M sodium dihydrogen phosphate (pH = 6.72) and acetonitrile (83:17, *v/v*) at a flow rate of 1.0 mL/min.

### 3.7. Congo Red Test

The Congo red method was performed to determine the triple helix structure of purified polysaccharides with slight modifications [33]. In brief, 2 mL polysaccharide solution (1 mg/mL) was mixed thoroughly with 2 mL of Congo red reagent (100 μMmol/L). Then, the NaOH solution (1 M) was gradually added to the mixture to establish a series of NaOH concentrations (0, 0.1, 0.2, 0.3, 0.4, and 0.5 M). After 15 min, the maximum absorbance of the mixture was measured with UV spectrophotometer (UV2600) at a wavelength of 400–600 nm.

### 3.8. SEM Analysis

The morphological characteristics of polysaccharides were observed using a scanning electron microscope (Phenom Prox, Eindhoven, The Netherlands). The dried powder samples were mounted on a metal stub using a conductive adhesive. The images were taken at 6000× magnification.

### 3.9. Antioxidant Activity Analysis

#### 3.9.1. Evaluation of DPPH Radical Scavenging Activity

The DPPH radical scavenging capacities of purified polysaccharides were measured according to the previously reported method with some modification [24]. A total of 2 mL of 0.04 mg/mL fresh-prepared DPPH-ethanol solution was mixed with 2 mL of the polysaccharides (0, 0.05, 0.10, 0.20, 0.40, 0.60, 0.80, and 1.00 mg/mL concentration). The reaction mixture was stirred and immediately incubated in the dark at room temperature for 30 min. The absorbance of the solution (A_s_) was determined at 517 nm. Also, the absorbance of the DPPH solution (A_b_) (2 mL) with 2 mL of ethanol, and the absorbance of polysaccharides sample solution (A_c_) (2 mL) with 2 mL of ethanol was measured as described before. Ascorbic acid was used as a positive control. The DPPH radical-scavenging activity of the samples was calculated as follows:DPPH scavenging rate (%) = [1 − (A_s_ − A_c_)]/A_b_ × 100

#### 3.9.2. Evaluation of Superoxide Anion Radical Scavenging Activity

The superoxide anion radical scavenging capacities were measured by the pyrogallol autoxidation method [34]. A total of 0.2 mL of (50 mmo1/L, pH = 8.0) pyrogallol solution with 0.2 mL of the polysaccharides samples (0, 0.05, 0.10, 0.20, 0.40, 0.60, 0.80, and 1.00 mg/mL concentration), and 2.8 mL of Tris-HCl solution (0.1 M, pH = 8.0) were incubated at 25 °C for 20 min, then the absorbance values (A_t_) of the mixture were measured at 320 nm using V_c_ as a positive control. The absorbance (A_0_) of the identical solution, which was prepared as the blank in exactly the same manner with the exception of the sample solution, which was replaced by distilled water, was recorded. The superoxide anion radical scavenging activity of the samples was calculated as follows:Scavenging rate (%) = (1 − A_t_/A_0_) × 100

#### 3.9.3. Evaluation of ABTS Radical Scavenging Activity

ABTS scavenging activity of polysaccharides samples was measured as described earlier [35]. Briefly, 10 mL of (2.45 mM) potassium persulfate solution and 10 mL of (7 mM) ABTS solution were separately prepared then mixed. The mixture was allowed to stand in the dark for about 12 h at room temperature before use. Different concentrations of sample solution (0.4 mL; 0, 0.05, 0.10, 0.20, 0.40, 0.60, 0.80, and 1.00 mg/mL) were added to 3 mL of the above activated pregenerated ABTS solution. This solution was diluted with 95% ethanol to yield an absorbance of 0.70 ± 0.02 at 734 nm. The mixture was kept in the dark for 10 min at room temperature, and the absorbance (A_a_) was measured at 734 nm. The absorbance (A_d_) of the identical solution, which was prepared as the blank in exactly the same manner with the exception of the sample solution, which was replaced by distilled water, was recorded. V_c_ was used as a positive control. The ABTS radical scavenging activity of the samples was calculated as follows:Scavenging rate (%) = ((A_d_ − A_a_)/A_d_) × 100

#### 3.9.4. Evaluation of Hydrogen Peroxide Hemolysis

H_2_O_2_-induced hemolysis assay was assessed by a previously reported method with minor modification [36]. Briefly, ICR (Institute of Cancer Research) mouse eyeball blood collected, using heparin-anticoagulant tubes. After centrifugation at 3000 rpm at 4 °C the plasma was removed, and the red blood cells (RBC) were collected and washed with buffered saline three times to prepare the 0.5% RBC suspension. A total of 4 mL of 0.5% RBC suspension was mixed with 4 mL of *E. prolifera* polysaccharide (at each concentration of 0, 0.05, 0.10, 0.20, 0.40, 0.60, 0.80 and 1.00 mg/mL) and incubated with 2 mL of 50 mmol/L H_2_O_2_ at 37 °C for 1–3 h. After incubation, the mixtures were centrifuged for 5 min at 3000 rpm at 4 °C. The absorbance of the supernatant was measured at 415 nm. Whereas A_e_ was defined as the absorbance of the sample, A_f_ was defined as the absorbance of the blank group in the absence of sample and H_2_O_2_, and the A_g_ was defined the absorbance of distilled water. The V_c_ was used as a positive control. The hemolysis inhibitory activity was calculated as follows:Hemolysis inhibitory rate (%) = ((A_e_ − A_f_) /(A_g_ − A_f_)) ×100

### 3.10. Antitumor Activity In Vitro

The antiproliferation activity of the crude polysaccharides, PAP-1, PAP-2, PAP-3, and PAP-4, was tested in vitro on MGC-803 (Human gastric cancer cells), HepG2 (Human liver cancer cells), T24 (Human bladder cancer cells), Bel-7402 (Human liver cancer cells), and AG1522 (Normal human foreskin fibroblast cells) and was evaluated by the MTT-based colorimetric method [37]. All cells were cultured in medium DMEM at a of density 4 × 10^4^ cells/cm^2^. Cells were incubated in a humidified atmosphere of 5% CO_2_ at 37 °C. A total of 0.04 mg/mL of polysaccharide samples was added to the test well. Continuing incubation for 48 h at 37 °C and in 5% CO_2_ atmosphere, 10 μL of MTT solution was added into each well and the cultures were incubated further for 4 h. After removal of the supernatant, DMSO (100 μL) was added to dissolve the formazan crystals. The absorbance was measured at 570/630 nm. The inhibition rate was calculated as follows:Inhibition rate (%) = (1 − A_sample_/A_control_) × 100

### 3.11. Statistic Analysis

All experiments were repeated three times, and the data were expressed as the mean ± standard deviation (SD). The Statistical analysis of differences were performed using a one-way analysis of variance (ANOVA). Differences were considered significant at *p* < 0.05.

## 4. Conclusions

In this study, crude polysaccharide (AP) was extracted from *E. prolifera* through alkaline extraction method, then isolated and purified to obtain four fractions (PAP-1, PAP-2, PAP-3, and PAP-4). Their molecular weights were 3.44 × 10^4^, 6.42 × 10^4^, 1.20 × 10^5^, and 4.82 × 10^4^ Da, respectively. HPLC analysis showed that PAP-1 and PAP-2 were mainly composed of galacturonic acid, while PAP-3, and PAP-4 were mainly composed of rhamnose. Congo red test showed that no triple helical structure was detected in the four polysaccharides. Finally, antioxidant assays showed that these polysaccharides had scavenging ability on DPPH, superoxide anion radical, ABTS radical assays, and an inhibitory effect on H_2_O_2_ induced red blood cell hemolysis. PAP-1 showed stronger antioxidant activity in vitro. Remarkably, we expect that this work would serve as a basis for future research on alkali-extracted polysaccharides from *E. prolifera*, and promote the exploitation and utilization of *E. prolifera*.

## Figures and Tables

**Figure 1 marinedrugs-18-00552-f001:**
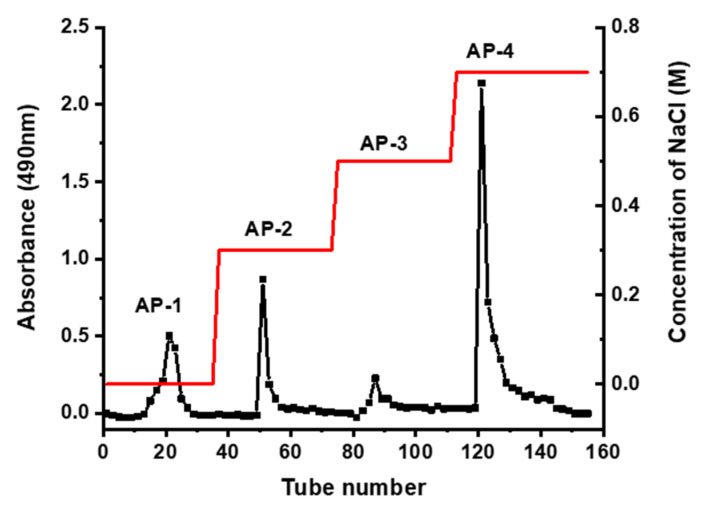
Elution profile of AP by DEAE-52 cellulose column chromatography.

**Figure 2 marinedrugs-18-00552-f002:**
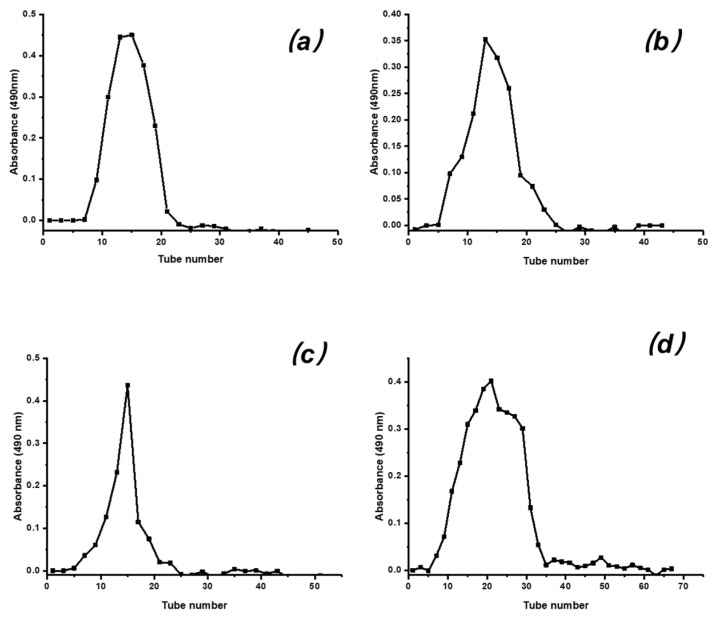
Elution profile of PAP by gel filtration chromatography on Sephadex G-100 (**a**): PAP-1, (**b**): PAP-2, (**c**): PAP-3, (**d**): PAP-4.

**Figure 3 marinedrugs-18-00552-f003:**
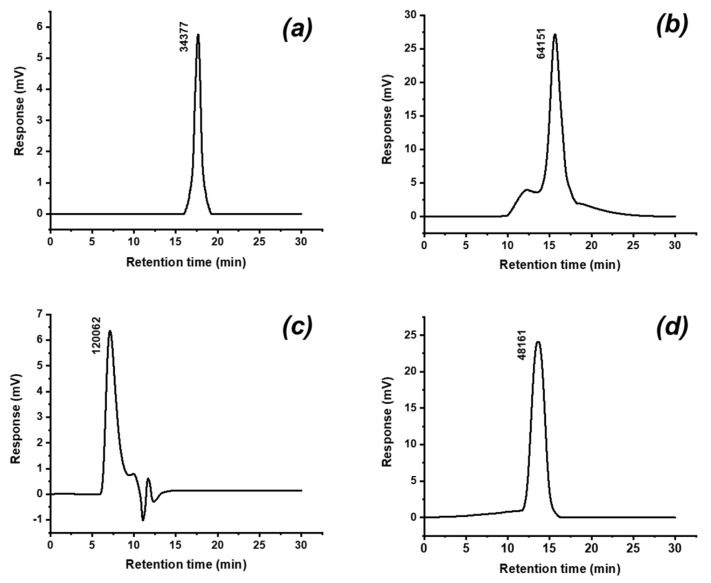
HPGPC chromatograms of purified polysaccharides (**a**): PAP-1, (**b**): PAP-2, (**c**): PAP-3, (**d**): PAP-4.

**Figure 4 marinedrugs-18-00552-f004:**
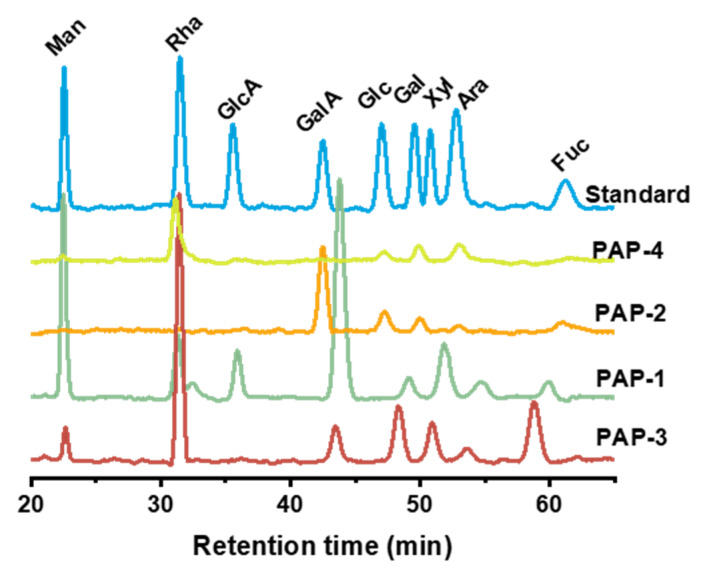
The HPLC chromatogram of standard monosaccharides and purified polysaccharides.

**Figure 5 marinedrugs-18-00552-f005:**
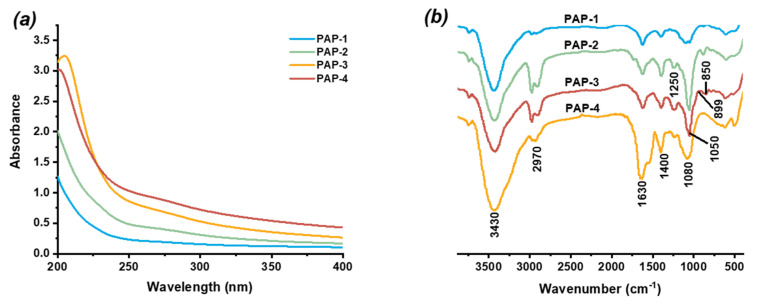
UV (**a**) and FT-IR (**b**) spectra of the purified polysaccharides.

**Figure 6 marinedrugs-18-00552-f006:**
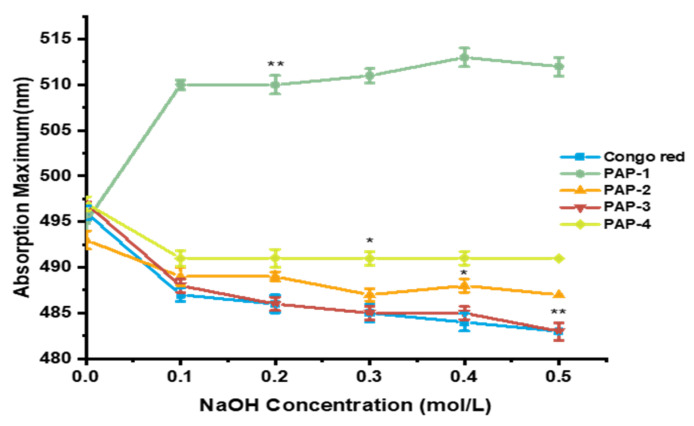
The UV absorbance spectrum of purified polysaccharides in Congo red assay. Each value represents the mean ± SD (*n* = 3); * *p* < 0.05, ** *p* < 0.01.

**Figure 7 marinedrugs-18-00552-f007:**
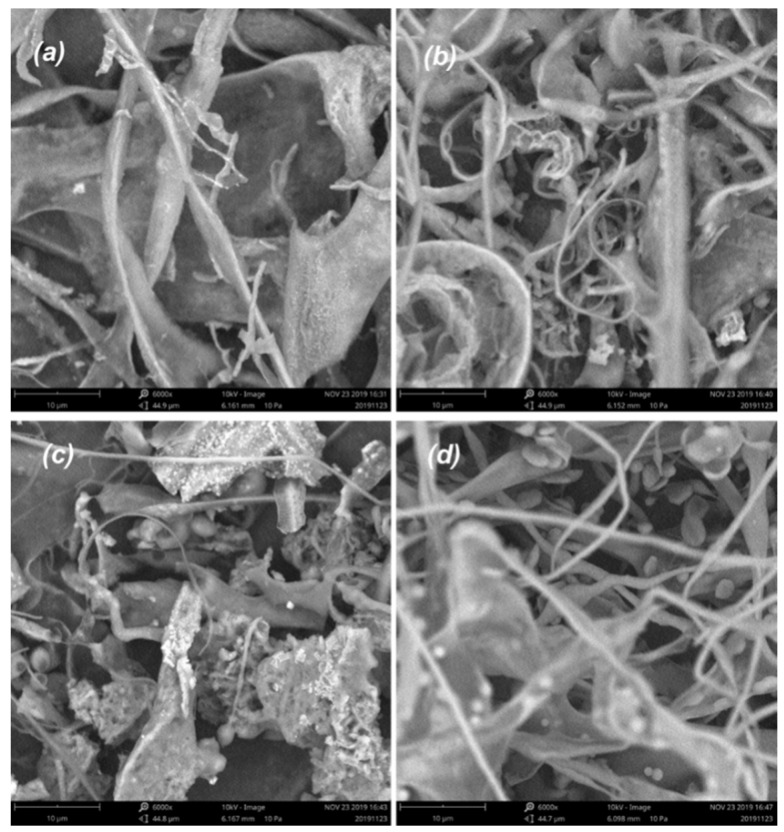
SEM micrographs (×6000) of polysaccharides, (**a**): PAP-1, (**b**): PAP-2, (**c**): PAP-3, (**d**): PAP-4.

**Figure 8 marinedrugs-18-00552-f008:**
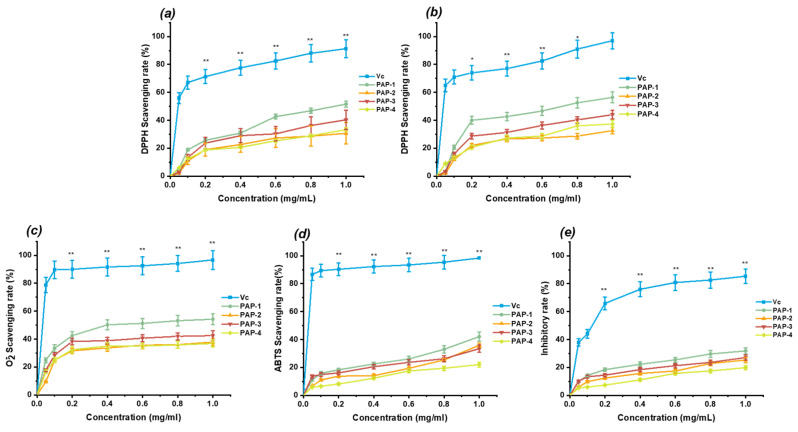
Scavenging effects of polysaccharides: (**a**) DPPH assay in 15 min, (**b**) DPPH assay in 30 min, (**c**) Superoxide radicals assay, (**d**) ABTS radicals assay, (**e**) Inhibitory on H_2_O_2_ Induces Red Blood Cell Hemolysis assay. Each value represents the mean ± SD (*n* = 3), * *p* < 0.05, ** *p* < 0.01.

**Table 1 marinedrugs-18-00552-t001:** Monosaccharides composition of *E. prolifera*.

Monosaccharide Ratio (%)	Samples			
	PAP-1	PAP-2	PAP-3	PAP-4
Mannose	21.28	n.d.	4.38	2.59
Rhamnose	7.80	n.d.	43.49	63.29
Glucuronic acid	7.19	n.d.	n.d.	n.d.
Galacturonic acid	42.10	63.91	8.21	n.d.
Glucose	n.d.	12.53	13.80	5.95
Galactose	3.17	7.67	n.d.	12.80
Xylose	11.11	n.d.	8.66	15.37
Arabinose	2.58	4.18	3.33	n.d.
Fucose	4.77	11.71	18.13	n.d.

n.d.- not defected.

**Table 2 marinedrugs-18-00552-t002:** Inhibition rate at concentration of 0.04 mg/mL (%).

	MGC-803	T24	HepG2	Bel-7402	AG1522
PAP-1	25.69	29.57	26.13	25.15	4.46
PAP-2	24.53	14.81	12.67	22.97	2.19
PAP-3	14.70	20.61	22.81	26.93	2.83
PAP-4	25.62	18.97	23.12	21.17	3.34
AP-1	21.59	23.10	19.86	17.39	3.51
AP-2	18.96	17.55	12.16	17.98	2.01
AP-3	15.65	18.63	22.35	23.05	2.54
AP-4	11.69	12.67	14.81	20.51	3.21
AP	17.73	17.78	26.12	17.89	2.89
DOX	100	100	100	100	68.31

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
