# Peer review of "Isolation, Characterization and Bioactive Properties of Alkali-Extracted Polysaccharides from Enteromorpha prolifera"

_marinedrugs, 2020, doi:10.3390/md18110552_

Round 1

Reviewer 1 Report

The presented review is very interesting and innovative; however, the authors used only transformed cell lines; in this respect, an incubation with, for example, normal human fibroblasts, is necessary. Authors should indicate in the Materials and Methods the name of cell lines used and the cell culture medium in which they grow. Besides, since polysaccharides can often induce osmotic changes, which may be responsible for the cytotoxic activity on cell lines (especially at high concentrations), the authors must perform a hemolysis assay. Moreover, the authors should express the concentrations in mg/ml, rather than in micromolar. Finally, the conclusions of the work are rather reductive and strangely no comment is made regarding the cytotoxic activity of APs

Lane 139 remove the period and add the comma and T from upper case to lower case.

Reviewer 2 Report

The paper entitled ‘Isolation, Characterization and Bioactive Properties of  Alkali-extracted Polysaccharides from Enteromorpha  prolifera’ presents interesting studies results. The manuscript is well prepared, the used methods are adequate and clearly described. Neverheless, some points have to be improved:

  1. authors decided to incubate dpph-extract solution by 30min and after this time the absorbance was measured. In my view, the time is too long because even weak antioxidant can scavenge free radicals after The adequate time will be max 15min.
  2. There is no information about statistical analysis. This point is crucial for evaluation the studies results. Please provide statistical analysis and all figures and tables the analysis should include the analysis (standard deviation)
  3. Discussion is insufficient. In my opinion, the Authors provided only studies results without detail discussion.

Round 2

Reviewer 1 Report

I thank the authors for answering my questions.

Author Response

Thank you for your reply.